# KGYM: A Platform and Dataset to Benchmark Large Language Models on Linux Kernel Crash Resolution

**Alex Mathai** [1*], **Chenxi Huang** [2*], **Petros Maniatis** [3]

**Aleksandr Nogikh** [4], **Franjo Ivančić** [4], **Junfeng Yang** [1] and **Baishakhi Ray** [1]

[1]Columbia University, [2] University of Minnesota, [3] Google Deepmind, [4] Google
{alexmathai, junfeng, rayb}@cs.columbia.edu
{maniatis, nogikh, ivancic}@google.com
{huan2677}@umn.edu

## Abstract

Large Language Models (LLMs) are consistently improving at increasingly realistic software engineering (SE) tasks. In real-world software stacks, significant SE effort is spent developing foundational system software like the Linux kernel. Unlike application-level software, a systems codebase like Linux is multilingual (low-level C/Assembly/Bash/Rust); gigantic (>20 million lines); critical (impacting billions of devices worldwide), and highly concurrent (involving complex multi-threading). To evaluate if machine learning (ML) models are useful while developing such large-scale systems-level software, we introduce KGYM (a platform) and KBENCHSYZ (a dataset). The KGYM[2] platform provides a SE environment for large-scale experiments on the Linux kernel, including compiling and running kernels in parallel across several virtual machines, detecting operations and crashes, inspecting logs, and querying and patching the code base. We use KGYM to facilitate evaluation on KBENCHSYZ, a crash resolution benchmark drawn from real-world Linux kernel bugs. An example bug in KBENCHSYZ contains crashing stack traces, a bug-reproducer file, a developer-written fix, and other associated data. To understand current performance, we conduct baseline experiments by prompting LLMs to resolve Linux kernel crashes. Our initial evaluations reveal that the best performing LLM achieves 0.72% and 5.38% in the unassisted and assisted (i.e., buggy files disclosed to the model) settings, respectively. These results highlight the need for further research to enhance model performance in SE tasks. Improving performance on KBENCHSYZ requires models to master new learning skills, including understanding the cause of crashes and repairing faults, writing memory-safe and hardware-aware code, and understanding concurrency. As a result, this work opens up multiple avenues of research at the intersection of machine learning and systems software.

## 1 Introduction

In recent years, there has been significant progress in using code LLMs (like CodeWhisperer [Amazon, 2023] and CoPilot [GitHub, 2021]) in all stages of the software cycle, including development, debugging, and testing. Despite being trained on large and complex open-source projects, LLMs are

---

[*]Denotes equal contribution
[2]https://github.com/Alex-Mathai-98/kGym-Kernel-Playground

38th Conference on Neural Information Processing Systems (NeurIPS 2024) Track on Datasets and Benchmarks.

often benchmarked on test sets like EvalPlus [Liu et al., 2023a], HumanEval [Chen et al., 2021], and APPS [Hendrycks et al., 2021] which are about[3] to get saturated [Ott et al., 2022] . While useful, these benchmarks represent "green-field" SE by isolating coding to the task of solving programming puzzles. Unfortunately, such puzzles do not reflect the intricacies involved in everyday reasoning and solving of complex bugs in production-ready software.

Hence, newly introduced benchmarks (like SWE-Bench [Jimenez et al., 2024]) try to bridge the gap between existing tasks and realistic SE in "brown-field" environments, where LLM assistants edit, debug, and test production-ready software. Such benchmarks capture a more realistic SE setting: given a software repository, a natural-language (NL) description of a problem or feature request, and a set of held-out executable test cases, edit the repository so that the test cases pass.

Our work moves one step further along the same trajectory, by introducing a drastically more challenging SE benchmark for future assistants. Specifically, we target *crash resolution in the Linux kernel* [Lin]: given a state of the Linux codebase, a crash report, and the crash-inducing input, the target is to repair the codebase such that the input no longer triggers a crash. To that effect, we build an execution environment, KGYM, and corresponding benchmark, KBENCHSYZ.

**Why Linux?** The Linux Kernel spans over 20M lines of code spread across 50k files. It has been in open-source development for decades and is deployed on billions of devices worldwide, including cloud infrastructures, desktops, and over three billion active Android devices [And]. Although the criticality of Linux itself justifies a benchmark built around it, KBENCHSYZ also tests LLM assistants on new and generalizable SE skills beyond what is available today:

- `Low level`: Linux is a systems codebase written in a mixture of C, Assembly (for multiple hardware architectures, like x86, ARM, etc.), Bash, and Rust, sometimes intermingled in the same file (e.g., in Assembly embedded in C). As a result, the implementation must be hardware-aware and memory-safe, in contrast to userspace code (often hardware-agnostic) and code in managed languages such as Python (the runtime abstracts away memory and hardware details).
- `Concurrent`: Linux code is highly concurrent and non-deterministic, with many kernel bugs caused by hard-to-reproduce thread interleavings, leading to deadlocks, race conditions, and atomicity violations ("Heisenbugs"). To resolve such bugs, the model must be able to learn and reason about the different interleaving schedules across concurrent threads. Moreover, a corresponding benchmark platform must work with *flaky* test oracles—the bug is sometimes observed, but not always—unlike existing benchmarks, which rely on deterministic oracles.
- `Ambiguous Intent`: Unlike application-level SE tasks that start with an NL description of a problem, the root cause in a crash report is often unknown, hard to reproduce, and must be identified before it can be resolved. This makes for a challenging task, both in terms of ambiguity and the underlying dependence on myriad behaviors of the complex kernel.
- `Decentralized Development`: Linux development is highly decentralized; a recent version (v6.3) saw contributions from $\sim$ 2k developers, with 513k lines deleted and 644k lines added [RV6]. Such decentralized development is managed by splitting the kernel into subsystems, each with head maintainers. Consequently, each subsystem has unique coding conventions, including custom memory allocators, complex data structures, and specific coding templates.

KBENCHSYZ consists of 279 Linux-kernel bug-resolution samples. Each consists of (i) a commit-id that specifies a kernel code-base exhibiting the bug crash; (ii) a crash report containing one (or more) crash stack traces; (iii) a reproducer (i.e., a crashing test input program); (iv) a developer-written and vetted patch that, when applied to the kernel code-base, fixes the root cause of the crash and results in an operational kernel; and (v) compilation and execution configuration files for the above. Additionally, it provides detailed email discussions between kernel developers leading to a bug's resolution. The samples are diverse, covering multiple critical subsystems, exhibiting various crash types, and requiring fixes from a single line to many lines across multiple functions and files.

**KBENCHC**. We have also curated and released a larger dataset of 504 Linux-kernel bugs. KBENCHC and KBENCHSYZ differ in the artifact used to reproduce the kernel bugs. While KBENCHSYZ uses a *syz* reproducer file (explained in Section 2) to reproduce the bugs, KBENCHC uses a C code snippet that has been derived (translated) from the original *syz* reproducer.

For the remainder of this paper however, we describe and report results on KBENCHSYZ.

---

[3]https://evalplus.github.io/leaderboard.html

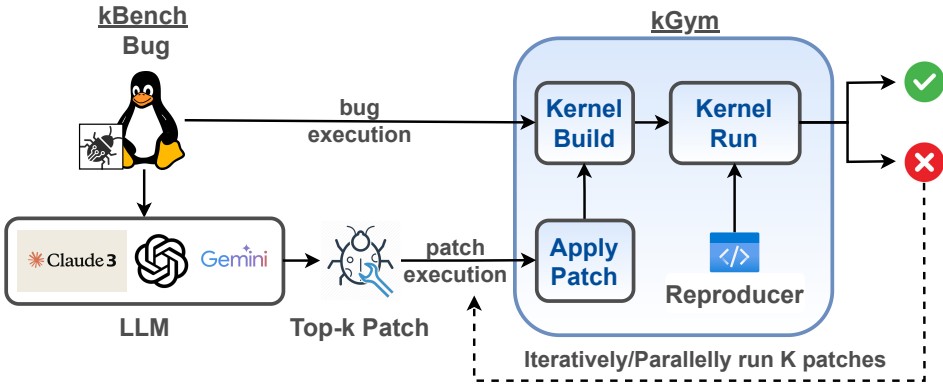

Figure 1: κGYM Pipeline. Input to κGYM is a κBENCHSYZ bug consisting of a kernel crash and a crash reproducer file. To reproduce the bug, κGYM compiles the buggy kernel version and runs the reproducer file. Next, the LLM is prompted with the kernel bug (along with the crash trace) to generate potential patch(es). Each code patch is given to κGYM, which then applies the patch to the buggy kernel version, compiles the entire kernel, and subsequently executes a reproducer file to check if the bug has been successfully resolved.

**κGYM** provides an execution platform for ML-assisted SE to address challenges in κBENCHSYZ. It is scalable, user-friendly, and capable of (a) compiling hundreds of Linux kernel versions, (b) applying patches to buggy kernels, and (c) executing bug reproducers to either replicate a Linux kernel bug or confirm crash resolution after a patch. A sample end-to-end run of κGYM is shown in Figure 1. As depicted, κGYM facilitates a typical debug-patch-test cycle, where given a crash report an LLM is called to generate a code patch (or Top-K patches). κGYM then applies the patch to the buggy kernel, runs the reproducer, and returns results: which can be another crash or a successful resolution. Key features of κGYM for κBENCHSYZ include parallel execution across VMs and replicated test execution to manage non-determinism. Equipped with parallel execution, κGYM can run hundreds to thousands of iterations of this loop within a day with limited resources, thus supporting further research in AI-assisted SE and low-level systems software.

Using κGYM, we first reproduced all 279 bugs in κBENCHSYZ and then used LLMs in the pipelines to fix them. In this process, we ran over 17k kernel jobs using both open-source and state-of-the-art LLMs. In both RAG-based assisted (5.38%) and unassisted (0.72%) settings, our results show that even the best LLMs perform poorly on Linux kernel crash resolution, suggesting that κBENCHSYZ is poised to establish itself as the next frontier benchmark for LLM-assisted SE.

In what follows we list the different kernel bug components, provide details about κGYM, describe how we collected κBENCHSYZ, and present initial crash-resolution results using popular LLMs.

## 2   Background: Continuous Testing via Syzkaller in Syzbot

To enhance Linux kernel security, the security community has developed numerous fuzzing tools over the past decade [Syz, Tri, Schumilo et al., 2017, Kim et al., 2020]. These tools automatically mutate and prioritize inputs to test the kernel, aiming to find bugs that developers can eventually resolve. We choose Syzkaller [Syz] to construct κBENCHSYZ as it is a widely-used open-source testing service for the Linux Kernel, where developers post, discuss, and fix kernel bugs. To date, more than 5k Syzkaller-detected kernel bugs have been reported and fixed, far surpassing the total bugs detected in the two decades before Syzkaller's inception in 2016 [CVE].

**Syzkaller** generates inputs resembling user-space programs by mutating a domain-specific language (DSL) called *syz* and can optionally translate this into a C program. Thus, the `input` to a kernel is itself a `program` containing a sequence of up to 10 Linux kernel system calls. The specifics of the *syz* DSL and how Syzkaller mutates the input are beyond the scope of this paper; what is relevant is that the input to each κBENCHSYZ sample is a user-space program produced by Syzkaller, which we also refer to as the `Reproducer`. For the bugs in κBENCHC, the input to each sample is the C program translation of the Syzkaller user-space program. Unlike the *syz* program which is relatively short, the C translation can be arbitrarily long as it directly depends on the output of the translation procedure.

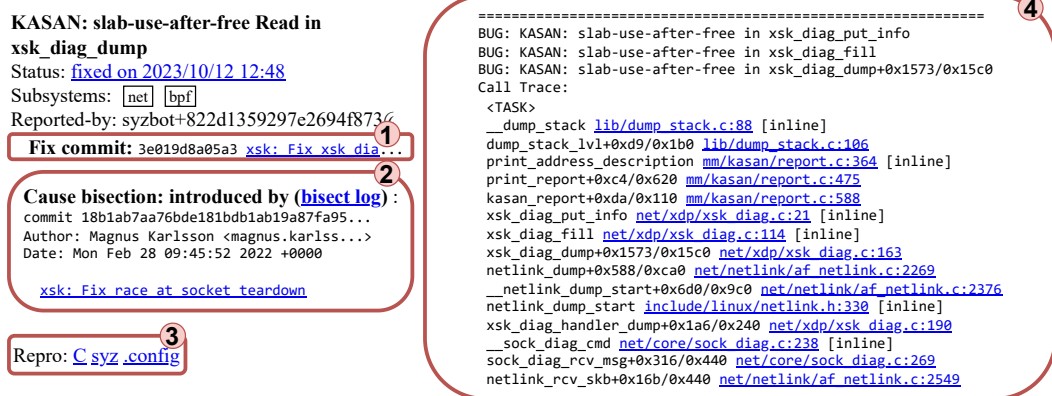

Figure 2: A sample kernel bug from Syzkaller [Bug]

**Syzbot** is an open-source platform that continuously runs Syzkaller on numerous kernels spanning multiple versions, architectures, and branches; testing them against various fault detectors. These detectors range from simple ones that detect kernel deadlocks or crashes (a.k.a., kernel "panic", when the kernel reaches an irrecoverable fault state) to complex ones looking for high-priority assertion violations. Many such detectors are called *sanitizers*[Stepanov and Serebryany, 2015, Con, Serebryany et al., 2012, Add], which typically look for concurrency and memory-safety issues. For instance, KASAN, the Kernel Address Sanitizer[Serebryany et al., 2012], detects memory corruption such as out-of-bounds reads and use-after-free accesses. Whenever a fault detector is triggered during a Syzkaller run, a kernel crash report is posted on a public Syzbot site (Figure 2). Kernel developers discuss the report, propose fixes, and the crash is considered resolved when the reproducer no longer triggers the crash and a maintainer accepts the fix.

We collect KBENCHSYZ samples (as shown in Figure 2) from the reported and fixed bugs on Syzbot. More specifically, for each bug, we collect

  i. Commit$_{bug}$: the specific kernel commit id exhibiting the crash.
  ii. Config : a file that specifies options and flags needed to correctly compile the Linux kernel.
  iii. Reproducer: the bug reproducer (③ in Figure 2) that triggers the crash.
  iv. Commit$_{fix}$ and Fix (gold fix): fix commit id and developer patch that resolves the bug (①).
  v. Crash$_{bug}$: the crash report and stack traces generated at the commit id Commit$_{bug}$ (④).
  vi. Bisect: a cause-bisection commit identifying the first commit that exposed the bug (available for $\sim 20\%$ of bugs) (②).
  vii. Email: email discussions of developers about the bug. This is included as auxiliary information for bug localization, explanation, and repair research.

## 3   KGYM: A Scalable Platform for Kernel Bug Reproduction and Resolution

KGYM is a scalable, flexible, extensible, and user-friendly platform for research using LLM-assisted tools on Linux Kernel SE problems. Below, we list the inputs to KGYM and the different actions that KGYM provides to a user (here "user" can refer to an AI agent) to apply patches, build kernels, and run reproducers (Figure 1). We highlight two important functionalities of KGYM, Kbuilder and Kreproducer. For an in-depth explanation of KGYM's architecture, see Appendix 3.

**Inputs:** From the features discussed in Section 2, we only need the commit id, the Config, and the Reproducer to reproduce a bug using KGYM. Additionally, we provide a crash report to the LLM to help it generate a patch. Using these inputs, KGYM can perform the list of actions mentioned below.

**Build:** For kernel crash resolution, we must first enable the building of kernels at specified commit ids. We provide a kernel-building API supported by Kbuilder, that focuses on compiling a kernel based on user specifications. These include a *git-url*, a *commit-id* (e.g., Commit$_{bug}$), a *kernel-config* (Config), a *compiler*, a *linker*, a *hardware architecture* (currently amd64), and a *userspace image* (options: buildroot, debian-bullseye, debian-buster, debian-stretch).

**Reproduce-Bug:** Once the user builds a kernel, the next step is to run the `Reproducer` to generate the crash report. We provide a bug-reproducing API supported by Kreproducer. This API requires (i) a pre-compiled disk image (from `Build`) and (ii) a `Reproducer` file. Kreproducer launches a VM with the image, monitors the reproducer's execution, and collects kernel panic information if a crash occurs. Thus, using `Reproduce-Bug`, we can generate and collect the crash report for the Linux kernel bug. However, since many bugs are non-deterministic, running the reproducer once may not suffice. A `Parallel-Reproduce` action launches multiple VMs to run `Reproduce-Bug` in parallel, increasing the chances of reproducing the kernel crash.

**Retrieve-File:** After obtaining the crash report, the next steps are to (i) retrieve relevant code files from the Linux codebase, (ii) inspect these files, and (iii) suggest a patch. The `Retrieve-File` action fetches files from the Linux codebase at a specific commit-id by checking out the correct commit and retrieving the specified files.

**Patch:** The input prompt to the LLM is constructed using the crash report and retrieved files. The LLM then generates a fix, which can be applied to the codebase at the specified commit-id. To check for crash resolution, the user must first apply the fix, recompile the Linux kernel, and then re-run the `Reproducer`. The `Patch` action facilitates this by taking a *patch* argument specified in the `git diff` format in addition to all the `Build` action arguments. The `Patch` action applies the patch and compiles the kernel. The user can then use this compiled kernel with the `Reproduce-Bug` action. If the `Reproducer` does not crash the kernel within 10 minutes, the bug is considered resolved.

**Kernel-Log:** For future works that monitor the Linux kernel environment, we provide the `Kernel-log` action. When invoked, `Kernel-log` downloads the Kernel's ring buffer (`dmesg`[4] output) for inspection after applying and running a patch. Analyzing kernel log changes is challenging due to its verbosity, often containing hundreds of thousands of lines. Although we believe this log will become crucial in kernel crash resolution, we leave this as a future research area.

## 4 KBENCHSYZ

We use the KGYM system explained in Section 3 to curate KBENCHSYZ, a dataset of Linux kernel bugs and fixes. We then use this dataset to benchmark the efficacy of state-of-the-art LLMs in solving bugs in production-ready software. In what follows, we explain how we derive a gold standard subset of bugs from the Syzkaller dataset and then delve into the characteristics of the benchmark itself.

**Notion of a Fix:** We follow Syzkaller and deem a patch as a valid bug fix if, upon application of the patch, the kernel remains functional without a crash, after executing the `Reproducer`.

**Retrieving Kernel Versions to Apply Fixes** ($Commit_{parent}$): For many bugs, there can be thousands of commits between $Commit_{bug}$ and $Commit_{fix}$ because patches for old bugs are often submitted to the current latest kernel version. Hence, to verify if a patch successfully resolves a crash, we must first compute the last commit before $Commit_{fix}$ where the bug is still reproducible. This is $Commit_{parent}$, which is the parent commit immediately before $Commit_{fix}$ in the git tree.

**Filtering a Gold Standard:** For each bug, we collect $Commit_{bug}$, Config, Reproducer, $Commit_{fix}$, Fix, and $Commit_{parent}$ (where we will apply the fix). We filter the bugs using three criteria: (1) The kernel crashes when running `Reproducer` on $Commit_{bug}$, (2) The kernel crashes when running `Reproducer` on $Commit_{parent}$, and (3) The kernel does not crash when running `Reproducer` on $Commit_{fix}$. These checks ensure each data point is a valid reproducible bug with a demonstrable fix.

**Experiment Caveat:** In Section 5, we perform all crash resolution experiments on $Commit_{parent}$ to allow for a qualitative comparison of the LLM's suggested patch against the actual Fix. Therefore, we provide the crash report generated at $Commit_{parent}$ as part of the input prompt to the LLM. Using KGYM, we run every bug in KBENCHSYZ and collect $Crash_{parent}$, the crash report observed when running the `Reproducer` on $Commit_{parent}$. Consequently, each data point in KBENCHSYZ is characterized by a seven-tuple: ($Commit_{bug}$, Config, Reproducer, $Commit_{fix}$, $Commit_{parent}$, $Crash_{parent}$, Fix). It is important to note that crash resolution can still be attempted on $Commit_{bug}$. But due to the thousands of commits between $Commit_{bug}$ and $Commit_{parent}$, the correct solution for $Commit_{bug}$ may differ vastly from the Fix. In the following section, we perform some quantitative studies of KBENCHSYZ to better understand the characteristics of this benchmark.

---

[4]https://en.wikipedia.org/wiki/Dmesg

## 4.1 Characteristics of the KBENCHSYZ

Table 1: Kernel Versions

| Kernel Version | Bugs |
|---|---|
| 4.x.x (2015 onwards) | 26 |
| 5.x.x (2019 onwards) | 141 |
| 6.x.x (2022 onwards) | 112 |

Table 2: Fix Types

| Fix Type | Bugs |
|---|---|
| Single Line | 33 |
| Single Function but Multiline | 145 |
| Multi Function but Single File | 57 |
| Multi Files | 44 |

Table 3: Line/File Statistics

| Data Type | Avg / Max |
|---|---|
| GF Lines Changed | 14.27 / 147 |
| GF Files Changed | 1.28 / 7 |
| $Crash_{parent}$ Lines | 84.3 / 624 |

**Kernel Versions:** The Linux kernel continuously evolves with contributions from thousands of developers worldwide, resulting in major releases every 3 to 5 years. Additionally, each major version is supported with updates for almost 10 years after the release. Capturing this diversity is important in our dataset. Table 1 shows the distribution of kernel versions in KBENCHSYZ, which includes a range of versions from the past 10 years (versions 4 to 6).

**Fix types:** To measure performance on varied types of fixes, it is important to ensure fix diversity in the KBENCHSYZ. Hence, we consciously include kernel bugs with varied fix sizes—from smaller single-line fixes to larger multi-file fixes. In Table 2, we show a detailed distribution of our dataset.

**Line statistics:** In addition to fix types, it is important to consider the line/file-level statistics of the gold fixes (GF) and the kernel crashes in the dataset. In Table. 3, we show the distribution of these statistics across the Dataset. As shown, the average lines changed in a GF is 14.27 (maximum of 147), and the average files changed in a GF is 1.28 (maximum of 7). Similarly, we observe that the kernel crash report is very verbose with an average of 84.3 and a maximum of 624 lines respectively.

**Fix Distribution Over Time:** To better understand the temporal distribution of fixes in KBENCHSYZ, we study the number of Linux bug fixes accepted each year from 2018 to 2023. As shown in Table 4, KBENCHSYZ has temporal diversity with many bugs from recent as well as past years.

**Git Tree:** Syzkaller has discovered bugs in numerous git trees of Linux. However, for the initial version of KBENCHSYZ, we stick to the mainline git tree and will eventually expand to other trees.

**Subsystem Distribution:** The Linux kernel is broken down into individual subsystems to streamline maintenance and development. Each kernel subsystem is actively maintained by a unique team of kernel experts. As a result, KBENCHSYZ should ideally have diverse bugs spanning multiple subsystems. As shown in Figure 3, KBENCHSYZ has bugs from 72 subsystems with net (network), usb and fs (filesystem) being the three biggest categories.

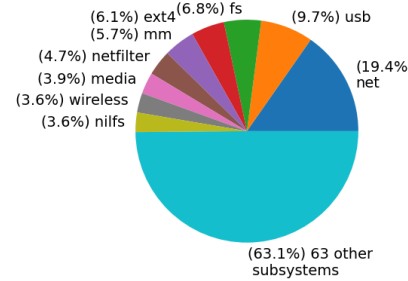

Figure 3: Subsystem Distribution

Table 4: Fix Distribution Over Time

| Year | Number of Fixes |
|---|---|
| 2023 | 79 |
| 2022 | 82 |
| 2021 | 34 |
| 2020 | 44 |
| 2019 | 20 |
| 2018 | 20 |

## 5 Experiments

We conduct extensive baseline experiments to benchmark state-of-the-art LLMs on KBENCHSYZ. We describe how we construct the input prompt, list the open and closed LLMs used, outline the two testing settings, and present a qualitative and quantitative analysis of the results.

## 5.1 Models

**Closed LLMs:** We conduct experiments on state-of-the-art closed LLMs like GPT-3.5 Turbo, GPT-4 Turbo, Claude-3 Sonnet, and Gemini-1.5 Pro. For GPT-3.5 Turbo, we use a maximum context length of 16k tokens. For more powerful models like GPT-4 Turbo, Gemini-1.5 Pro, and Claude-3 Sonnet, we use a maximum context size of 50k tokens to stay within budget constraints.

**Open LLMs:** We also experiment with the Llama series of open-source instruction-tuned LLMs like Code Llama-7b-Instruct, Code Llama-13b-Instruct, Code Llama-34b-Instruct, and Llama-3-8B-Instruct. To stay within resource constraints, we restrict ourselves to a maximum context length of 16k tokens.

## 5.2 Input Prompt

To generate viable kernel patches, we provide meaningful context in the LLM's input prompt. For each bug in KBENCHSYZ, the prompt includes $Crash_{parent}$ and relevant C files (Section A.3). Since Linux Kernel files can be thousands of lines long, prompts often exceed the maximum context lengths of LLMs. Therefore, we run experiments on smaller subsets of KBENCHSYZ, detailed in Section 5.3.

## 5.3 Evaluation Settings

An important part of the input prompt is a set of C files relevant to the crash report. Given the Linux kernel's vast size, selecting the most relevant files is challenging. Following SWE-Bench [Jimenez et al., 2024], we use a retrieval-based system for this task and evaluate each LLM in two settings:(1) oracle retrieval and (2) sparse retrieval. It is important to note that in both settings, we limit the kernel crash report to a maximum of 10k tokens to keep enough space for the relevant C files.

**Oracle Retrieval:** In this setting, we parse the actual developer `Fix` and collect the modified files. Each modified file is included in the prompt, and the LLM is asked to generate a patch for these files. This assisted setting makes the task easier, but we are forced to skip the bug if all Oracle files do not fit into a single prompt. This reduces the number of bugs to 117 for models with a 16k context size (e.g., GPT-3.5 Turbo and Llama models) and 228 for models with a 50k context size (e.g., GPT-4 Turbo, Claude-3 Sonnet, and Gemini-1.5 Pro).

**Sparse Retrieval:** In the unassisted setting, the bug is first localized to a set of C files before the LLM generates a patch. This localization can be done using many techniques. Dense retrieval mechanisms are ill-suited [Jimenez et al., 2024] because of the sheer scale of the Linux kernel ($> 20M$ lines and $> 50k$ files). Hence, using $Crash_{Parent}$ as the key, we adopt a sparse retrieval method like BM25 to retrieve the top 3 files to modify. Once we get the top 3 files, we add as many files as possible to the input prompt without exceeding the context limit. However, we intentionally skip a kernel bug if we cannot fit a single file. For models with a context length of 16k, the number of bugs is reduced to 227, and for a longer context length of 50k, we get 275 bugs. Please refer to Table 10 in the Appendix for a tabular view of the model variants against their respective KBENCHSYZ subsets.

**BM25 Efficacy:** To evaluate BM25, we compare its retrieval predictions against the set of Oracle files. As shown in Table 5, as we retrieve more predictions from BM25 by increasing $K$ from 3 to 20, the number of samples for which BM25 returns a superset of the oracle files increases from $1.76\%$ to $9.69\%$ for a 16k context length and from $2.91\%$ to $10.54\%$ for a 50k context length. As evident from Table 5, there is a lot of scope to improve bug localization using a given kernel crash report. However,

Table 5: BM25 Recall for different values of Top-K and context lengths. All, Any and None denote complete, partial, and no overlap with oracle files respectively.

| | BM25 Recall | |
| | 16K | 50K |
| --- | --- | --- |
| Top K | All / Any / None | All / Any / None |
| 3 | 1.76 / 0.00 / 98.24 | 2.91 / 0.00 / 97.09 |
| 5 | 3.96 / 0.44 / 95.6 | 5.10 / 0.36 / 94.54 |
| 10 | 6.61 / 0.00 / 93.39 | 7.64 / 0.00 / 92.36 |
| 20 | 9.69 / 0.44 / 89.87 | 10.54 / 0.36 / 89.10 |

these results are unsurprising, because if we set $K$ to 3 and assume a single Oracle file, the probability of correctly including the Oracle file in 3 random choices from the 50k files in the Linux kernel is 0.006. Thus, in contrast to random guessing, BM25 does a reasonable job.

## 5.4 Quantitative Analysis of Patch Generation

**Querying LLMs:** To stay within budget and API constraints, we query each LLM differently. For GPT-3.5 Turbo and GPT-4 Turbo APIs, we ask for the top-10 likely patches. By extracting 10 outputs (instead of 1), the total cost increases by only 20-30% as the long input context exhausts most of the budget. The Gemini-1.5 Pro API does not provide a parameter for multiple outputs, but as it is currently free to use, we query Gemini 10 times with the same input tokens. There is also no such parameter for the paid Claude-3 Sonnet API, so we conduct experiments with a single output to limit costs. As such, the Claude API metrics should likely improve if 10 outputs are considered.

**Compute:** To run the crash resolution experiments using κGYM at scale, we employ 11 VMs hosted on Google Cloud. Each VM is a `c2-standard-30` Google Compute Engine (GCE) instance.

Table 6: Patch Application and Bug Solve Rates using state-of-the-art LLMs. CL stands for CodeLlama and L3 stands for LLama-3. All % numbers are calculated for the entire 279 bugs in κBENCH-SYZ. We ran over 17,000 kernel jobs using κGYM to quantify these results.

| Top-N | Patch Results | GPT-3.5 Turbo (1, 10) | CL 7b (1) | CL 13b (1) | CL 34b (1) | L3 8B (1) | GPT-4 Turbo (1, 10) | Claude 3 Sonnet [5] (1) | Gemini 1.5 Pro (1, 10) |
|---|---|---|---|---|---|---|---|---|---|
| Oracle | **Apply %** | (1.43, **15.41**) | 9.68 | 0.72 | 15.41 | 0.36 | (20.07, **56.99**) | 27.60 | (22.22, 45.52) |
| | **Solve %** | (0, **1.08**) | 0 | 0 | 0 | 0 | (1.08, **5.38**) | 1.79 | (0.72, 3.58) |
| BM25 | **Apply %** | (13.26, **40.86**) | 20.79 | 0.72 | 40.14 | 1.08 | (15.77, **55.20**) | 28.67 | (12.19, 24.37) |
| | **Solve %** | (0.36, **0.36**) | 0 | 0 | 0 | 0 | (0, **0.72**) | 0 | (0, 0) |

**Patch Application Rate:** As part of the input prompt, we ask the LLM to generate a git `diff` patch for κGYM to apply to the codebase. However, we observe that current state-of-the-art LLMs often struggle to generate *syntactically valid* patches with the correct `diff` structure. This issue has also been noted in other works like SWE-Bench [Jimenez et al., 2024]. Table 6 shows the patch application rate (Apply %) for each LLM in both the Oracle and BM25 settings to illustrate the prevalence of this problem. Amongst the 50k context models (GPT-4 Turbo, Claude-3 Sonnet, and Gemini-1.5 Pro), GPT-4 Turbo achieves the highest application rate as it generates well-formed patches for more than half the bugs in both the Oracle (56.99%) and BM25 settings (55.2%). For the remaining 16k context models, we notice the highest Apply % for GPT-3.5 Turbo in both settings (15.41% and 40.86%).

**Bug Solve Rate:** In addition to the patch application rate, we also measure % of *semantically valid* patches, i.e., the % of bugs solved when successfully applying the patch (Solve %). Amongst the 50k context models, GPT-4 Turbo has the highest solve rate of 5.38%, solving 15 bugs in the Oracle setting. However, in the BM25 setting, due to poor retrieval performance, only GPT-4 Turbo has a non-zero solve rate of 0.72% indicating that more research is needed to improve kernel bug localization and resolution. For the 16k context models, GPT-3.5 Turbo has the highest solve rates of 1.08% and 0.36% in the Oracle and BM25 settings respectively. Unfortunately, the solve rates for all the Llama models are 0% in all scenarios.

**Union:** Upon inspecting all the correct LLM patches, we note 29 unique bug ids from a total of 36 solved bugs. Hence, combining the patches from all the models results in a solve rate of 10.39%.

Overall, we observe that state-of-the-art LLMs struggle to effectively resolve Linux kernel bugs due to the sheer complexity and scale of the problem. As a result, we believe that there is a lot of scope for research to make LLMs effective in this domain. In the following section, we will qualitatively analyze an example patch from GPT-4 Turbo and compare this to the actual `Fix`.

## 5.5 Qualitative Analysis of Patch Generation

Figure 4 shows an example of a `memory leak` bug in κBENCHSYZ. The crash report (left) includes a stack trace with `cinergyt2_frontend_attach` highlighted in red, which is the buggy function that is modified in the `Fix`. On the right, we compare the actual `Fix` by a developer with a successful patch suggested by GPT-4 Turbo. The model's patch correctly localizes the bug but is less nuanced and safe than the developer's solution. The developer's fix ensures memory safety and follows coding conventions, while the model's patch uses `kfree`, which can cause issues if the memory was not

---

[5]Top-10 results for Claude-3-Sonnet were skipped due to budget constraints

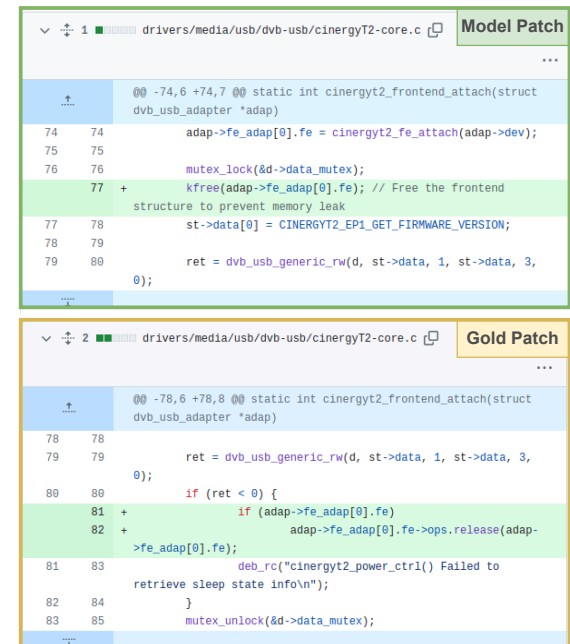

Figure 4: A sample bug patch using GPT-4 Turbo. The left figure shows a stack trace with the buggy function highlighted in red. The right compares a successfully generated patch by GPT-4 Turbo vs a human developer. The developer solution first confirms that `adap->fe_adap[0].fe` is not `null` and then uses the function pointer field `ops.release` to deallocate the structure safely using a custom memory deallocator. In contrast, the model uses `kfree` in the generated patch to deallocate the object which implicitly assumes that the object was allocated memory using `kmalloc`.

allocated with `kmalloc`. Despite its shortcomings, the model's patch can expedite debugging by highlighting the root cause, guiding the developer in composing a more accurate fix, thereby speeding up kernel crash resolution.

## 6 Related Work

**Code Modeling and ML for SE.** Recent advancements in code LMs have made program synthesis a reality [Guo et al., 2022, Ahmad et al., 2021, Wang et al., 2021, Feng et al., 2020]. Many efforts have also scaled these advancements to build models that show amazing code comprehension and completion capabilities [lla, Rozière et al., 2024, Nijkamp et al., 2023a,b, Fried et al., 2023, Chen et al., 2021]. Subsequently, many works have adapted code LMs to assist in various SE tasks like testing [Xia et al., 2024, Wang et al., 2024, Kang et al., 2023], program repair [Dinh et al., 2023, Gao et al., 2022], commit generation [Liu et al., 2023b], and pull request reviews [Li et al., 2022]. Program repair is the closest research area to this work. However, previous works have not explored program repair in the context of massive systems-level repositories. We believe this is partly because performing large-scale experiments on these codebases is very challenging. Hence, we hope that κGYM will spur research at the intersection of ML and systems-level code.

**Benchmarking.** The most commonly evaluated application of Code LLMs is code generation. As a result, there are a plethora of code completion benchmarks. Most benchmarks including HumanEval [Chen et al., 2021] and others [CodeGeeX, 2022, Austin et al., 2021, Athiwaratkun et al., 2023, Cassano et al., 2023, Hendrycks et al., 2021, Lu et al., 2021, Puri et al., 2021, Clement et al., 2021, Ding et al., 2023a, Wang et al., 2023, Lu et al., 2022] mainly assess code completion by providing in-file context, i.e., the LLM prompts only contain code from a single file. More recent works have introduced tougher repository-level benchmarks [Shrivastava et al., 2023, Ding et al., 2022, Pei et al., 2023, Zhang et al., 2023, Ding et al., 2023b, Jimenez et al., 2024]. Among these, SWE-bench (Jimenez et al. [2024]) is the closest related work as it concentrates on repository-level program repair.

However, unlike SWE-bench, KBENCHSYZ focuses on low-level systems code, not generic userspace code like Python libraries. Additionally, a sample KBENCHSYZ problem has a code context scale that is 50 times the size of the largest SWE-bench instance. Hence, we believe that progress made on KBENCHSYZ would reflect advancements in the real-world crash resolution capabilities of ML models.

## 7 Limitations

**Time Intensive Experiments**. Due to the large size of the Linux kernel, compilation and linking of $\sim$ 50k files takes a significant amount of time. Using a `c2-standard-30` machine, the compilation process takes 15 to 20 minutes. After this, if a patch is correct, the reproducer runs for 10 minutes, and if incorrect, the kernel crashes within a couple of minutes. Thus, the total feedback time, after patch application, ranges from 17 to 30 minutes. As such, running kernel experiments is time intensive.

**Single Test Reproducer**. As KGYM builds upon Syzkaller, it uses a single `Reproducer` to check if the crash goes away. As a result, it is possible for an LLM to generate a patch that may work for crash resolution but may significantly alter code functionality. Thus after the reproducer check, extensive testing maybe required to ensure that the kernel functions correctly. This can be done by using tools like Linux Testing Project[6] or Linux Kernel Selftests[7].

## 8 Conclusion

In this work, we introduce KBENCHSYZ (and KBENCHC), a new challenging SE benchmark aimed at Linux kernel crash resolution. To effectively experiment on KBENCHSYZ, we also introduce KGYM, a platform to execute large-scale kernel experiments. To interact with KGYM, we also provide few simple APIs. Using KGYM, we run over 17k kernel jobs to report our initial baseline results that indicate poor performance even when using state-of-the-art LLMs. Thus, we conclude that there is adequate scope for research to improve crash resolution performance in massive production-ready systems-level codebases. We hope that by introducing KBENCHSYZ and KGYM, we spur more efforts that lower the barrier of entry to research at the intersection of machine learning and system software.

## 9 Acknowledgements

This work was partly supported by multiple Google Cyber NYC awards, a Columbia SEAS/EVPR Stimulus award and the following NSF awards - IIS 2221943, CCF 2313055, CCF 1845893, CCF 2107405.

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

# A    Appendix / supplemental material

## A.1    κGYM: Background and Architecture

### A.1.1    Background: Syzkaller

Despite Syzkaller's many features, in practice, it is challenging to conveniently leverage Syzkaller to perform large-scale experiments on the Linux kernel. As a result, Syzkaller is often out of reach for the average code ML researcher but is routinely used by experienced kernel developers.

**Syz-build and Syz-crush:** With this in mind, we implement κGYM- a platform for ML-for-code researchers that is scalable and easy to use. We allow researchers to compile, execute, and monitor Linux kernels at scale by invoking a few simple APIs! To realize this goal, we first isolate and re-use some components of Syzkaller to build the basic blocks of κGYM. As shown in Figure 5, the two main components in κGYM are `Kbuilder` and `Kreproducer`. Kbuilder is designated the task of compiling a kernel when provided with a kernel config file and a specific Git commit id. When executing Kbuilder, we invoke Syzkaller's *syz-build* utility - a robust tool developed in the Go language to compile various Linux kernel versions. Kreproducer on the other hand, executes a set of inputs (i.e., a reproducer file) on a pre-compiled kernel image. When we call Kreproducer, we internally invoke Syzkaller's *syz-crush* module to run either C programs or Syzkaller's domain-specific language (DSL) to reproduce identified bugs.

**Scaling Kernel Compilation and Test Execution:** The main advantage of using the κGYM system is that it can massively parallelize both the compilation of kernels as well as the execution of reproducer files. In our everyday experiments, we seamlessly run κGYM on 10 VMs, achieving a speed of 720 kernel compilations and reproducer executions within 24 hours. The ability to perform kernel experiments at this scale makes it practical and feasible for researchers to conduct tangible research at the intersection of LLMs and kernel bugs. In the following section, we delve into the fine architectural details of κGYM that make this possible.

### A.1.2    Architecture

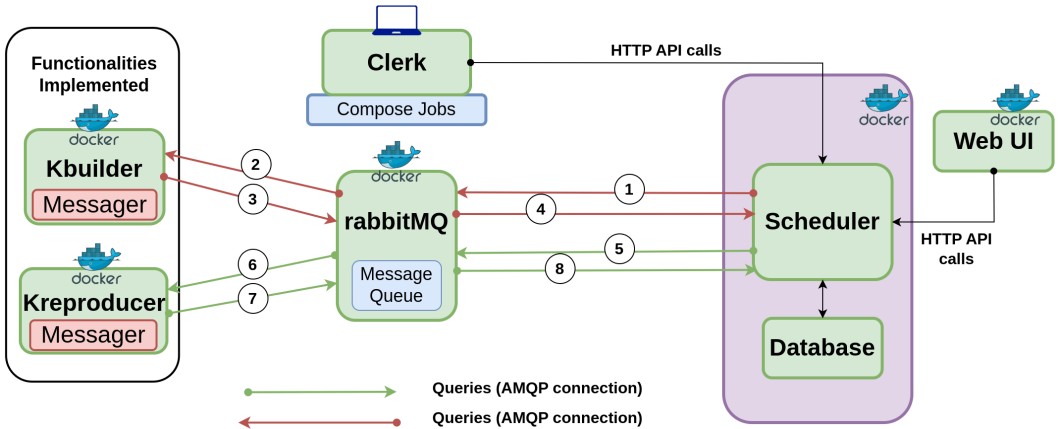

Figure 5: The κGYM Architecture

**κGYM:** κGYM is a scalable, flexible, extensible, and user-friendly platform for experimentation on the Linux Kernel. In what follows, we expand on each component of the κGYM Architecture and then summarize the merits of κGYM.

**Kbuilder:** Kbuilder purely focuses on the task of compiling a kernel according to the user's specifications. These specifications include (1) *git-url* - a URL to the git tree of the kernel, (2) *commit-id* - a specific git commit id, (3) *kernel-config* - the set of config values to use when compiling the codebase, (4) *user-img* - the userspace image to run the compiled kernel on (currently we give four options - `buildroot`, `debian-bullseye`, `debian-buster` and `debian-stretch`) , (5) *compiler* - the choice of either *gcc* or *clang* to compile the kernel, (6) *linker* - the choice of either *ld* or *ld.lld*

to link the modules, (7) *arch* - the architecture of the compiled kernel (currently we only support `amd64`) and (8) *patch* - an optional patch specified in a git diff format.

Using the above inputs, Kbuilder clones the git repository, performs a git checkout to the specified commit id, applies the patch if provided, compiles the kernel with *syz-build* using the compiler, linker and kernel config, places the kernel in the userspace image and finally uploads the entire disk-image to Google Cloud Storage. Note, as it takes a long time to even clone the Linux kernel (10 to 20 minutes), we optimize this step by caching Linux codebases from different git trees, thus allowing us to start the build process from the git checkout step.

**Kreproducer:** After compiling the Linux kernel and storing the disk image, we then execute the reproducer file using Kreproducer. As Syzkaller runs all of its fuzzing operations on GCE (google cloud engine) instances, we try to replicate this reproduction environment to maximize our chances of reproducing a bug. Hence, Kreproducer uses a pre-complied disk image (either from Kbuilder or otherwise) to launch a GCE instance and runs a reproducer file that internally invokes a series of system calls on the kernel. Kreproducer then monitors and collects important information during the execution. If the reproducer file crashes the instance, Kreproducer will collect kernel panic information from the serial port output of the instance. However, if the reproducer file does not crash the kernel, the reproducer continues to run until the maximum time elapses (10 minutes by default). Hence using Kreproducer we can effectively determine if the bug has been resolved or if the bug persists.

**Scheduler:** One of the main reasons why KGYM is scalable is because of the architectural design of the scheduler. When a user submits a batch job containing hundreds of kernel compilations and executions, the scheduler inspects each job and delegates parts of each job to either the Kbuilder or Kreproducer. Additionally, as multiple Kbuilders/Kreproducers can be hosted on separate VMs, the scheduler can coordinate the execution of multiple jobs at a time. The scheduler keeps track of each job and its execution state in a lightweight SQLite3 database. We also provide an easy web UI that queries this database to provide real-time updates on each job.

**Clerk:** To make scheduling of jobs even easier, we offer *Clerk* - a client-side library that exposes many APIs for kernel building and reproducer file execution. Each API internally invokes the scheduler to run different kinds of jobs. Armed with KGYM and Clerk, code LLM researchers can now schedule kernel experiments with just a few lines of python code!

**KGYM Workflow:** We complete our explanation of KGYM with a dry-run of a representative kernel job. In this example, we assume that the kernel job involves both a kernel compilation and a reproducer file execution. As shown in Figure 5, we first issue this job using the Clerk library. The scheduler inspects the incoming job and notes two sequential and dependent steps - (a) building a kernel and (b) running a reproducer on the built kernel. To complete the first step, the scheduler issues a Kbuilder job using the message broker RabbitMQ (arrow ①). RabbitMQ then finds an available Kbuilder VM and issues this new job to the running Kbuilder (arrow ②). The Kbuilder accepts all the corresponding arguments, builds the kernel, and uploads the disk image to Google Cloud Storage. It then notifies the scheduler that the build process is completed by sending a message using a custom-built library called *messager* (arrow ③ and arrow ④). Once the scheduler receives this message, it starts the second step by issuing a reproducer job (arrow ⑤) to RabbitMQ, which includes Kbuilder's output in the arguments. Like before, RabbitMQ finds an available Kreproducer VM and assigns this job to the running Kreproducer (arrow ⑥). Kreproducer consumes the corresponding arguments and runs the reproducer file on the kernel image. The reproducer runs until the kernel crashes or until the maximum allotted time. The job then finishes when Kreproducer communicates its results back to the scheduler (arrow ⑦ and arrow ⑧). Any KGYM user can easily monitor this multi-step process via our simple web UI interface.

**Design Rationale:** We arrived at this architectural design to make sure that KGYM is scalable and extensible. To scale KGYM, a user can simply increase the number of Kbuilder and Kreproducer VMs without changing any code implementation. Additionally, if a developer desires to extend KGYM, he/she can implement a new functionality (say KTask) and containerize it in a separate docker container. To exploit the benefits of KGYM, the developer can simply communicate with the scheduler (via rabbitMQ) using the *messager* communication-library.

## A.2 Bug Localization

Table 7: Bug Localization efficacy (complete overlap) of LLMs on Linux kernel bugs

| Model Top-N | | GPT-4 Turbo (10) | GPT-3.5 Turbo (10) | Claude-3 Sonnet (1) | Gemini-1.5 Pro (10) | All Llama Models (1) |
|---|---|---|---|---|---|---|
| Fix Type | Total Bugs | Oracle / BM25 | | | | |
| Single-Line | 33 | 3 / 0 | 6 / 0 | 4 / 0 | 7 / 0 | 0 / 0 |
| Single-Func | 145 | 19 / 3 | 35 / 2 | 45 / 6 | 44 / 6 | 0-2 / 0-2 |
| Multi-Func | 57 | 0 / 0 | 7 / 0 | 2 / 0 | 2 / 0 | 0 / 0 |
| Multi-File | 44 | 0 / 0 | 3 / 0 | 1 / 0 | 1 / 0 | 0 / 0 |
| Total | 279 | 22 / 3 | 51 / 2 | 52 / 6 | 54 / 6 | 0-2 / 0-2 |

In addition to evaluating LLM-generated patches, we also study the bug localization ability of LLMs when given a kernel crash report. For this study, we perform a post-facto analysis of the generated patches from our crash resolution experiments in Table 6. For every `git diff` generated by an LLM, we extract all the modified functions and create a list of tuples of the form (function name, file name). These tuples are also extracted for every bug's `Fix`. We then compute the overlap of both lists to measure the LLM's ability to localize bugs.

**Performance across models:** In Table 7, for every queried LLM, we depict the number of bug patches (in the Oracle and BM25 settings) where the patch tuples are a superset of the `Fix` tuples. As seen, in the Oracle setting, the best results are achieved by Gemini-1.5 Pro closely followed by Claude-3 Sonnet and GPT-3.5 Turbo. It is important to note that despite only taking Top-1 from Claude-3 Sonnet, its bug localization performance is almost as good as a Top-10 output from Gemini-1.5 Pro. In the BM25 setting, both Claude-3 Sonnet, as well as Gemini-1.5 Pro, achieve a full overlap for 6 bugs. This low performance can be mainly attributed to the poor localization results of BM25.

For the open-source Llama models, bug localization is still a challenge with 2 being the best metric across all the Llama models in both settings.

**Performance across Fix Types:** When comparing the performance of models across `Fix` types, we notice that the best performance across models is in the Single Function category. This implies that for most models, the LLM-generated patches modify functions that overlap with the buggy function of the `Fix`. We also notice poor performance for the Multi-Function (but single file) and Multi-file categories. Hence, the LLMs struggle to include all the functions modified in the `Fix` when the developer-written fixes are complicated and spread out.

Table 8: Bug Localization efficacy (**partial** overlap) of LLMs on Linux kernel bugs

| Model Top-N | GPT-4 Turbo (10) | GPT-3.5 Turbo (10) | Claude-3 Sonnet (1) | Gemini-1.5 Pro (10) | All Llama Models (1) |
|---|---|---|---|---|---|
| | Oracle / BM25 | | | | |
| Partial Overlap | 18 / 2 | 12 / 4 | 28 / 4 | 22 / 3 | 0-1 / 0 |
| Overlap % | 31.6 / 29.16 | 47.91 / 39.58 | 38.16 / 45.83 | 36.29 / 50 | 0-50 / 0 |

For completeness, in Table 8, we provide the number of patches that partially overlap with the actual `Fix`. Additionally, we also provide the overlap ratio (i.e., recall) to quantify the degree of overlap in these cases.

**Overlap of Fix functions with crash report:** It is important to quantify how much information LLMs can use from $Crash_{parent}$ to successfully localize the buggy functions modified in the `Fix`. For this, in Table 9, we depict the overlap of the functions mentioned in the crash report against those modified by the `Fix`. As shown, in both the BM25 and Oracle settings, less than 30% of the crash reports have textual references to all the functions modified in the `Fix` patch (i.e., less than

30% complete overlap). Additionally, in both settings, more than 50% of crash reports have no overlap with the functions modified in the Fix. This indicates that bug localization is indeed a very challenging problem in the Linux codebase. Given the absence of information in $Crash_{parent}$, we believe that more information needs to be extracted from the dynamic traces of the execution to perform bug localization.

Table 9: Overlap between $Crash_{Parent}$ and Fix

| Setting | Complete Overlap | Partial Overlap | No Overlap | Total |
|---------|------------------|-----------------|------------|-------|
| BM25    | 75               | 45              | 155        | 275   |
| Oracle  | 67               | 39              | 121        | 227   |

## A.3 Prompt Template

Models are prompted with the template below during the crash resolution experiments.

```
You will be provided with a partial code base and an issue statement
explaining a problem to resolve.
<issue>
{CRASH TEXT}
</issue>


[start of file_1]
{file_1 text}
[end of file_1]
[start of file_2]
{file_2 text}
[end of file_2]
....


Here is an example of a patch file. It consists of changes to the code
base. It specifies the file names, the line numbers of each change,
and the removed and added lines. A single patch file can contain
changes to multiple files.

<patch>
--- a/file.py
+++ b/file.py
@@ -1,27 +1,35 @@
def euclidean(a, b):
- while b:
- a, b = b, a % b
- return a
+ if b == 0:
+ return a
+ return euclidean(b, a % b)

def bresenham(x0, y0, x1, y1):
points = []
dx = abs(x1 - x0)
dy = abs(y1 - y0)
- sx = 1 if x0 < x1 else -1
- sy = 1 if y0 < y1 else -1
- err = dx - dy
+ x, y = x0, y0
+ sx = -1 if x0 > x1 else 1
```

```
+ sy = -1 if y0 > y1 else 1
- while True:
- points.append((x0, y0))
- if x0 == x1 and y0 == y1:
- break
- e2 = 2 * err
- if e2 > -dy:
+ if dx > dy:
+ err = dx / 2.0
+ while x != x1:
+ points.append((x, y))
 err -= dy
- x0 += sx
- if e2 < dx:
- err += dx
- y0 += sy
+ if err < 0:
+ y += sy
+ err += dx
+ x += sx
+ else:
+ err = dy / 2.0
+ while y != y1:
+ points.append((x, y))
+ err -= dx
+ if err < 0:
+ x += sx
+ err += dy
+ y += sy
+ points.append((x, y))
 return points
</patch>

I need you to solve the provided issue by generating a single patch file
that I can apply directly to this repository using git apply. Please
respond with a single patch file in the format shown above.
Respond below:
```

## A.4  Subset of κBenchSyz for every model

Table 10: For each LLM, the final subset of bugs from the κBenchSyz depends on the chosen retrieval method and the maximum allowed context length.

| All Bugs | Retrieval Method | Context Length | GPT-3.5 Turbo | GPT-4 Turbo | Claude-3 Sonnet | Gemini-1.5 Pro | Llama Models |
|---|---|---|---|---|---|---|---|
| 279 | BM25 | 16K | 227 | × | × | × | 227 |
| 279 | BM25 | 50K | × | 275 | 275 | 275 | × |
| 279 | Oracle | 16K | 117 | × | × | × | 117 |
| 279 | Oracle | 50K | × | 228 | 228 | 228 | × |

