# OpenReview forum: "kGym: A Platform and Dataset to Benchmark Large Language Models on Linux Kernel Crash Resolution"
_NeurIPS.cc/2024/Datasets_and_Benchmarks_Track — NeurIPS 2024 Track Datasets and Benchmarks Poster_

### Official Review · Reviewer_KpsL · 2024-07-16
**KGym Linux Kernel Crash Resolution**

**Rating:** 8
**Confidence:** 3
**Clarity:** Yes.

**Review:**

I enjoyed reading this paper. It is well-written, addresses a difficult problem, and presents a useful dataset, benchmark and experimental platform.

**Strengths:**

- Very difficult task. As the authors point out, many of the existing benchmarks for code generation are more about solving puzzles than about every day tasks. This benchmark addresses that problem.
- The KGym platform is a very useful tool to facilitate this type of research as building kernels can be cumbersome.

**Additional Feedback:**

Nice work. Thanks!

**Correctness:**

Yes, the dataset is constructed in a sound way and the experiments are appropriate.

**Documentation:**

At the moment, the benchmark and platform are only included as supplementary files. There is documentation available that describes how to execute the benchmark.

**Limitations:**

The limitations are only touched upon very briefly. The authors state that there are no negative societal impacts, but one of the potentially negative societal impacts is that a platform like KGym also facilitates malicious users by enabling them to test exploits on many different kernels in parallel.

**Opportunities For Improvement:**

- What is the purpose of Commit_fix? Since there is a reproducer file, any proposed fix can just be verified, right? Is there any point in comparing it with the developer's fix, especially since the fixes may be quite different because there may be many commits in between the bug and the fix?

**Relation To Prior Work:**

Yes.

**Summary And Contributions:**

The authors present KGym, a platform for large-scale experiments on the Linux kernel, and KBench, a dataset to of crashing stack traces together with their reproducer files and developer-written fixes. The authors point out that foundational system sofware is much more complex than application software due to its size, concurrency and critical nature. The best-performing LLM currently achieves 0.72% on the benchmark in the unassisted mode and 5.38% in the assisted mode.

---

> ### Author Rebuttal · Authors · 2024-08-15
>
> **Reviewer 2**
>
> We thank the reviewer for such positive feedback and greatly appreciate their views about the work.
>
> 1. **What is the purpose of commit_fix? Is there any point in comparing it with the developer's fix, especially since the fixes may be quite different because there may be many commits in between the bug and the fix?** - In our paper, we conduct bug resolution experiments on the “commit_parent” (the commit just before commit_fix). By doing so, we provide the LLM with the exact environment faced by the kernel developer. As a result, we can compare the model-generated fix against the developer fix.  This would not have been possible had we conducted experiments on commit_bug (the original commit where the bug was first found).
>
> 2. **Limitations** - Yes, we intend to expand on existing (but very nuanced) limitations in the conclusion when we get an additional page after acceptance.
>
> **Societal impacts**
> 1. We concur that Kgym inherits the pros and cons of any large-scale testing software. Any generic testing service (including KGym) can be used to perform large-scale software testing as well as large-scale software exploitation. Fortunately, based on our experiences with existing fuzzing tools such as AFL++ and Syzbot, these tools have been mostly used to generate a positive societal impact by enabling developers to find vulnerabilities and fix them. So we have reason to believe the same for KGym.
>
> **Open Source** - As mentioned previously, we intend to share the open-source version of the repo very shortly, and we are excited for people to try out our platform.

---

### Official Review · Reviewer_BMph · 2024-07-25
**A Benchmark that Focuses on an Area of Improvement for LLMs**

**Rating:** 8
**Confidence:** 4

**Review:**

Quality: an end-to-end benchmark and evaluation platform for a large and challenging software codebase. kGym can interact directly with LLMs and evaluate their solutions.

Clarity: the paper was well written and easy to follow. It explained basic concepts of kernels for those who are not experienced in OS could understand the paper.

Originality: in principle, it is similar to SWE-Bench, but focuses on a challenging codebase, Linux, with difficult crash bugs that are are difficult to debug.

Significance: the fact that most LLMs have low success rate on the benchmark, will make this benchmark an area of research for the research community to focus to find new paradigms to improve or find alternatives to LLMs.

**Strengths:**

- An evaluation platform that can interact with LLMs and enable researchers to improve or innovate new learning algorithms
- A benchmark that has crashes and solutions
- The evaluation platform is flexible and enables testing solutions from LLMs that do not necessarily match the gold fix

**Additional Feedback:**

- line 7: "ML" acronym used here for the first time. Suggest to replace it with "machine learning (ML)"
- Figure 1: there is a typo as there is a missing "c" in "Interatively".
- Line 191: What is the difference between "Config" and "Reproducer"? I feel we need a specific definition for each.
- Line 191: I suggest to specify within the seven-tuple, what is provided in the prompt
- Line 206: what is the difference between a fix and a gold fix (GF)?
- Line 229: "LLama" should be "Llama" (the standard spelling using by Meta is "Llama" and "Code Llama")
- Line 252: Please add citation for BM25. Please also briefly explain on what basis does BM25 extract files. Is it based on error logs in the crash?
- Table 5 Caption: Suggest to specify "overlap with oracle files"
- Line 300: "llama" should be "Llama"

**Clarity:**

- Paper is well written. Explains the background clearly to non-experts. Explains the different components and motivations behind them for both kGym and kBench.

**Correctness:**

- Evaluation methods seem to be done in a proper way

**Documentation:**

- Data collection: paper explained how it collected the fixes from Linux's GitHub issues and PRs
- Data organization: paper explained the different fields of the datasets
- Data maintenance: paper didn't mention if it is willing to maintain or update the dataset or kBench
- Ethical and responsible use: N/A

**Limitations:**

- The paper doesn't really have a Limitations section or paragraph. I suggest to discuss Limitations in the Conclusion
- The paper did mention the limitations of Retrieve-File, which is indeed a challenging task

**Opportunities For Improvement:**

- Consider using RAG to augment LLMs to obtain higher solution rates (similar to what we see on SWE-Bench leaderboard).
- Extend the evaluation for interaction mode: if a patch proposed by an LLM fails, provide the crash report to the LLM and ask it to provide a different solution or improve its proposed solution
- Please provide link for open sourced kBench and kGym
- Line 300: "Unfortunately, the solve rates for all the llama models are 0% in all scenarios": At the time of writing the paper, there were Llam3 70B and CodeLlama 70B models available. I think it would have been fair to evaluate them... as the other closed source models like Claude, GPT, and Gemini are most likely bigger in parameter size than the Llama models evaluated in the paper.

**Relation To Prior Work:**

- Yes. I believe the Related Works section was concise and the paper specified the closest work to it: SWE-Bench

**Summary And Contributions:**

This paper presents:
- kGym: a platform to apply and test fixes to Linux kernel crashes
- kBench: a set of Linux kernel crash description with their fixes

and evaluates it on existing LLMs, showing that LLMs have low solution rate to such benchmark and hence opens the door for researchers to investigate novel solutions to this challenging task.

---

> ### Author Rebuttal · Authors · 2024-08-15
>
> **Reviewer 1**
>
> Firstly, we would like to thank the reviewer for putting in time and effort to provide such detailed feedback.
>
> **Opportunities for Improvement**
>
>  1. **Consider RAG to augment LLMs** - Thank you for this suggestion! Indeed we are currently experimenting with advanced RAG-based and Agent-based techniques that have shown far better performance on SWE-bench-like tasks. We have already witnessed an increase in metrics due to this. However, as kGym intended to establish a new problem and introduce a new platform, this was out of the paper’s scope.
>
> 2. **Extend the evaluation for interaction mode** - Our latest internal version of kGym already supports this.
>
> 3. **Please provide a link to kBench and kGym** - Yes, this will happen very shortly! We are excited for people to use this platform!
>
> 4. **Comparing against larger Llama models** - We agree with the reviewer that this would have been a more fair comparison. However, due to computing and budget constraints, we chose to run smaller Llama versions locally in our compute clusters. In any case, we also attribute the lower performance to smaller parameter sizes rather than any Llama-specific problem.
>
> 5. **Limitations** - Yes, we intend to expand on existing (but very nuanced) limitations in the conclusion when we get an additional page after acceptance.
>
> **Additional Feedback.**
>
> 1. **Difference between config and reproducer** - As Linux is ~20M lines, there are numerous options to compile the codebase that switches on/off many features. To automate this compilation process, kernel developers provide a config file with thousands of flags. This is the “config”.  The reproducer on the other hand is a program that is run on the kernel numerous times to reproduce an existing bug. We will make sure to include this in the paper.
>
> 2. There is no difference between fix and gold fix.
>
> 3. Thank you for pointing out the spelling mistake as well as other issues like acronym usage, the correct way to write Llama, the BM25 citation, and appropriate table captions.

---

> > ### Comment · Reviewer_BMph · 2024-08-15
> >
> > Thanks. After reading the rebuttal I would like to keep my score.

---

### Official Review · Reviewer_kfnU · 2024-08-23
**A Testbed and Benchmark for LLM-assisted Linux Kernel Crash Resolution**

**Rating:** 8
**Confidence:** 4

**Review:**

Clarity: The paper has a clear narrative and is easy to follow. The authors do a commendable job motivating their work by explaining the high-level characteristics of the Linux kernel and the challenges associated with debugging it. They are transparent about almost every detail associated with data collection for KBENCH and the software engineering effort that went into developing KGYM. Table 1, 2, 3 and 4 along with Figure 3 aptly elucidate the data distribution in KBENCH. The authors were honest about the resource constraints they had to navigate and the impact of those constraints on the scope of this work. The only aspect where I would like to see more detail is the exact hardware specifications of the GCE virtual machines they utilized for running the KBENCH experiments on their selection of LLMs.

Quality: The authors pinpoint a worthwhile problem to address: State-of-the-art LLMs perform abysmally on repository-level program repair for Linux kernel crash resolution. They highlight this problem through a rigorously-designed set of experiments whose results are clearly and transparently communicated. Their selection of the LLMs represent a realistic baseline, given that it spans most of the prominent open and proprietary models at the time of writing. Their contribution of KBENCH and KGYM lowers the bar of entry to research on LLM-guided Linux kernel crash resolution, potentially catalyzing fruitful research efforts by the broader community.

Originality: I interpreted KGYM as a leaner version of the pre-existing Syzkaller tool. It seems to offer an API for a subset of the functionality afforded by Syzkaller. This does not take anything away from the utility and significance of KGYM. However, I cannot say the same about its originality. KBENCH on the other hand is a notable initiative towards a domain-specific dataset for the Linux kernel. I think it has the potential to be a valuable fine-tuning dataset to improve the reportedly lackluster performance of the SOTA LLMs on this task.

Significance: The Linux kernel is a codebase that powers millions of devices worldwide. Even though there is a large open-source community maintaining and developing it, it's massive size, multilingual nature, and non-deterministic dynamics pose challenges for debugging it. The authors present a software testbed to evaluate and potentially improve LLMs for this task, thus assisting human developers in bug detection and crash resolution. Consequently, I think the work is quite significant and the topic is aligned with this venue.

**Strengths:**

* KBENCH dataset is clearly described in every aspect: the curation and pre-processing steps; contents of a sample; origin of the data; dataset-level distribution of kernel versions, fix types, and line/file statistics, subsystems, and temporal characteristics.

* KGYM platform has the potential to a useful playground for LLM researchers who focus on Linux kernel crash resolution. It is scalable enough to deal with the non-deterministic nature of Linux kernel's thread scheduling routines. The architecture and usage of the tool is rigorously described in the manuscript. KGYM can serve as a platform for training an end-to-end bug detection and resolution system potentially powered by an arbitrary number of LLMs collaborating in an agentic setting. It's applicability to hundreds of variants of the Linux kernel is also noteworthy.

* The authors establish a realistic baseline of LLM performance on Linux kernel crash resolution by selecting a comprehensive set of SOTA LLMs at the time of writing.

* Sufficient literature review is conducted to contextualize the work and distinguish it from the pre-existing literature on the topic.

**Additional Feedback:**

Thank you for your submission NeurIPS 2024 Datasets and Benchmarks Track. Best of luck in your future research!

**Clarity:**

The paper has a clear narrative and is easy to follow. The authors do a commendable job motivating their work by explaining the high-level characteristics of the Linux kernel and the challenges associated with debugging it. They are transparent about almost every detail associated with data collection for KBENCH and the software engineering effort that went into developing KGYM. Table 1, 2, 3 and 4 along with Figure 3 aptly elucidate the data distribution in KBENCH. The authors were honest about the resource constraints they had to navigate and the impact of those constraints on the scope of this work. The only aspect where I would like to see more detail is the exact hardware specifications of the GCE virtual machines they utilized for running the KBENCH experiments on their selection of LLMs.

**Correctness:**

The dataset is constructed in a sound manner. The collection and pre-processing steps are clearly outlined in Section 2 and Section 4. Evaluation is rigorous in the sense that it benchmarks a comprehensive set of mainstream LLMs. However, it remains incomplete because the authors had to navigate considerable resource limitations due to long prompts they had to feed into proprietary LLMs from OpenAI, Google and Anthropic. Nevertheless, I maintain that the experiment design is logical, and the results are reasonably explained.

**Documentation:**

There is sufficient detail on data collection for the KBENCH dataset. The paper outlines a step-by-step roadmap for reproducing the benchmarking results. Associated data and codes are also shared with the reviewers as supplementary material even though they are yet to be publicized online.

**Ethics:**

I do not suspect any ethical concerns that warrant further discussion or review.

**Limitations:**

*The authors have addressed the limitations below:*

* The source code for the KGYM testbed and the KBENCH dataset is not yet open-sourced. Therefore, reproducing the reported results is not possible despite the detailed instructions provided in the paper.

* The KBENCH dataset has only 279 linux kernel bugs. A larger dataset is needed for improving LLM performance on Linux kernel crash resolution through various fine-tuning strategies.

*I encourage the authors to address the following limitations:*

* Please provide further detail on the exact hardware specifications of the Google Cloud VMs you used in your experiments. This will be needed to reproduce your results.

* Please consider offering a brief discussion what comes next. Now that the shortcomings of LLMs in kernel crash resolution are established, how can KBENCH and KGYM assist in addressing this problem? Which directions would you recommend for future LLM research leveraging these assets?

**Opportunities For Improvement:**

* As the authors mentioned, KBENCH currently has only 279 Linux kernel bugs. This dataset size might be sufficient for an evaluation suite. However, the impact of this work would be dramatically increased by expanding this dataset to allow for fine-tuning LLMs for Linux kernel crash resolution. I was happy to read that authors are planning future work in this direction.

* The authors mention that they used 11 Google Cloud VMs with 'c2-standard-30' configuration. It would be helpful to see a few sentences on what this entails. For instances, how many GPUs of which kind are allocated to these VMs? What is the memory capacity? How many CPU cores are available? These details will be helpful for reproducibility once the KGYM source code and the KBENCH is open-sourced.

* The authors are proposing a convincing argument for the necessity of further research on improving LLMs for Linux kernel crash resolution. I do understand that the scope of this paper ends at bringing the problem of under the spotlight. However, I would like to see a discussion on how a larger KBENCH can be utilized in combination with KGYM to improve the currently abysmal performance of SOTA LLMs on this task. Perhaps the authors can provide a vision or a roadmap for how their contribution could be utilized in fine-tuning efforts.

* I think the future work should consider a more sophisticated retrieval strategy for the assisted crash resolution. As mentioned, the Linux kernel has over 50,000 files and over 20 million lines of code. Perhaps a hierarchical retrieval strategy like GraphRAG (Edge et. al.) could allow for better retrieval performance. I do understand the resource limitations communicated in the manuscript. That said, BM25 retrieval performance reported in Table 5 calls for a solution to the subpar retrieval performance so that the assisted crash resolution can be more accurately evaluated.

**Relation To Prior Work:**

A rigorous literature review clearly comes across in the manuscript. The authors explore previous work on code LMs and their applications to program synthesis, code comprehension and completion, testing, commit generation, pull request reviews, and program repair. The authors state that their work fits into the program repair category. However, it differs from other work, particularly from SWE-bench (Jimenez et. al. [2024]), by targeting low-level systems code like the Linux kernel instead of high-level user-space codes such as Python libraries. They also claim that KBENCH code context scale is at least 50 times larger than that of largest SWE-bench sample.

**Summary And Contributions:**

Main contributions made in paper are a testbed for LLM-assisted Linux kernel crash resolution (KGYM) and a benchmark (KBENCH) for evaluating LLM-based methods on repository-level program repair on low-level codebases such as the Linux kernel.

*Following factors render KGYM a significant contribution:*

* It is compatible with hundreds of linux kernel versions
* It affords the functionality to apply patches to buggy kernel codes.
* It can run multiple VMs concurrently to execute bug reproducers and test LLM-generated patches that resolve the crash. This concurrency allows for bootstrapping the non-deterministic nature of kernel level thread scheduling, thus allowing for better uncertainty estimation around the crash resolution experiments.
* KGYM can run O(100) to O(1000) iterations per day, thereby allowing sufficient throughput for end-to-end iterative crash resolution.

*Following factors render KBENCH a significant contribution:*

* 279 Linux-kernel bug-fixes, each consisting a pointer to the codebase exhibiting the bug, crash report containing stack traces, developer-written kernel patch to fix the bug, compilation and execution files for reproducing the bug and confirming its resolution, as well as written developer correspondence associated with the bug and the fix.
* The samples in the benchmark span multiple crucial subsystems, various crash types, and single-file as well as multi-file crash fixes.

The authors utilize their KGYM tool to assess the performance of a comprehensive selection of state-of-the-art LLMs on their KBENCH benchmark. For this purpose, they run over 17 thousand kernel jobs. Their results serve as a whistleblower for the abysmal baseline performance produced by their selection of LLMs. Therefore, they call for further research in repository-level program repair that focuses on massive, multilingual, low-level codebases such as the Linux kernel.

---

> ### Author Rebuttal · Authors · 2024-08-25
>
> **Reviewer 3**
>
> We thank the reviewer for providing such a detailed review of the paper and for asking very nuanced and important questions. The effort put into this review is deeply appreciated.
>
> **Opportunities For Improvement**
>
> 1. **Insufficient dataset size** - The current 279 data points will likely prove inadequate when fine-tuning large language models. Hence we have already begun work to expand this dataset. We have to date collected around 523 data points from the mainline git tree of the Linux kernel. If we further consider other git trees of the Linux kernel, we can likely collect around 2x to 4x of this updated dataset size.
>
> 2. **GCE exact specifications** - It is important to note that as KGym primarily performs Linux kernel compilation and execution, its deployment does not need GPUs - instead it only requires fast multi-core machines with enough disk space. As per Google Cloud Platform documentation, the c2-standard-30 specification implies that the virtual machine has 30 vCPUs with 120 GB RAM. Calls to proprietary LLMs are made through standard APIs provided by Open AI, Anthropic, and Google. Local Llama models were run on our local clusters and the Llama-generated patch was provided to KGym as a “git-diff”.
>
> 3. **Our future roadmap** - Currently, we believe that the best roadmap is to develop a “Kernel Agent” capable of searching, editing, compiling, and testing solutions for Linux kernel bugs. Once we have a baseline kernel agent that solves ~30% to 40% of the bugs, we can generate agent trajectories for each solved bug and subsequently train a large language model on these agent trajectories. This is our current future roadmap.
>
> 4. **Better retrieval strategies** - We concur with the reviewer that there is a need for better retrieval strategies in the non-oracle setting. We also agree and have stated that the BM25 performance is abysmal and needs to be improved significantly. Thank you for suggesting the GraphRAG strategy - it does indeed look very promising for something at the Linux kernel scale.

---

### Decision · Program_Chairs · 2024-09-26

**Decision:**

Accept (Poster)

**Comment:**

I want to thank the authors for their submission to NeurIPS 2024 Datasets and Benchmarks Track. I agree with the reviewer's highlighted strengths:
* Important and challenging problem: Evaluating LLMs on Linux kernel crash resolution is more complex than typical application-level software tasks.
* Introducing kGym: A valuable platform for conducting large-scale experiments on the Linux kernel, including compiling and running kernels across virtual machines.
* Providing kBench: A dataset of 279 real-world Linux kernel bugs, including crash reports, bug reproducers, and developer-written fixes.
* Comprehensive evaluation: Benchmarking a wide range of state-of-the-art LLMs on the kBench dataset.
* Highlighting a significant performance gap: Current LLMs perform poorly on this task, with the best model achieving only 0.72% and 5.38% success rates in unassisted and assisted settings respectively.
* Clear presentation: The paper is well-written, easy to follow, and explains concepts clearly for non-experts in operating systems.

However, what I find missing yet critical is a demonstration of the usefulness of the dataset and the kGym environment. While I agree with the authors and believe it is very likely to be useful, the credibility of this work will be substantially improved if the authors show results on fine-tuned LLMs (with a larger dataset) and demonstrate much better performances than 0.72% and 5.38% using their environment and dataset.

Note from PCs: We disagree with the AC assessment.